# Self-regulating and self-oscillating metal-organic framework hybrid plasmonic metasurfaces

Hajar Amyar[1,2], Davide Raffaele Ceratti[1,3], Henri Benisty [2], Andrea Cattoni[4,6], Mondher Besbes[2] & Marco Faustini [1,5] ✉

Metal-organic frameworks (MOFs) offer remarkable chemical versatility, structural diversity, and, in some cases, stimuli-responsiveness. In the latter case, they typically rely on external inputs to trigger these changes. In contrast, living systems possess the ability to internally self-regulate and autonomously adapt their properties without external intervention, utilizing internal feedback mechanisms. To fill this gap, we develop a MOF-based metasurface that exhibits autonomous optical self-regulation, dynamically adjusting light absorption in response to varying incident light intensity. This device integrates colloidal MOFs with a plasmonic metasurface to create a thermo-optical negative feedback mechanism based on vapor sorption in and out of the colloidal MOF device. The self-regulation process is dynamic, leading each MOF/antenna unit to exhibit self-oscillatory behavior in the presence of a constant external energy input, analogous to a light-fueled nanoscale steam engine. This proof-of-concept highlights the potential of harnessing MOFs and sorption processes for designing metasurfaces for adaptable optical applications. It also represents a first step toward the design of materials integrating feedback mechanisms and internal clocks paving the way for a new generation of porous materials with life-like autonomy.

Autonomous self-adaptation and self-regulation are unique features that typically distinguish living from artificial materials. A key to autonomous behavior is the ability to integrate internal self-regulation and program dynamic actions over time, with spontaneous oscillations being a common example[1]. Biological self-regulation and self-oscillations are programmed at the molecular level and orchestrated through complex networks of chemical reactions and internal feedback mechanisms[2,3]. Developing artificial materials with life-like behaviors has long been a dream in materials science[4–6]. Artificial chemical systems exhibiting self-regulation[7] and, in some case self-oscillatory behavior have been synthesized and used to study out-of-equilibrium

reaction-diffusion systems[8,9] or induce spontaneous self-assembly[10,11], motion of hydrogels[12], or supramolecular systems[13,14] in liquid solution. Spontaneous oscillations have also been achieved in colloidal solutions through spontaneous aggregation and disaggregation mediated by chemical reactions[15,16] or plasmonic heating in liquid suspensions[17]. Recent astonishing examples of autonomous motions have been reported in devices made of liquid crystals or hydrogels for robotic applications[18–23]. Expanding the concepts of self-regulation to other families of materials, such as porous materials, could one day enable the design of programmable self-regulating porous devices capable of adapting autonomously to perturbations or be programmed in time[24].

[1]Sorbonne Université, CNRS, Laboratoire Chimie de la Matière Condensée de Paris (LCMCP), Paris, France. [2]Université Paris-Saclay, Institut d'Optique Graduate School, CNRS, Laboratoire Charles Fabry, Palaiseau, France. [3]PSL-University, CNRS-Chimie ParisTech, Institut de Recherche de Chimie Paris, Research Group of Physical Chemistry of Surfaces, Paris, France. [4]Centre de Nanosciences et de Nanotechnologies (C2N), CNRS UMR 9001, Université Paris-Saclay, Palaiseau, France. [5]Institut Universitaire de France (IUF), Paris, France. [6]Present address: Dipartimento di Fisica, Politecnico di Milano, Milano, Italy. ✉ e-mail: marco.faustini@sorbonne-universite.fr

This would open up completely new functionalities for a number of applications related to porous materials, such as (electro)catalysis, gas capture, thermal or optical management, and sensing[25,26]. Due to their chemical versatility and structural diversity, porous materials such as metal–organic frameworks (MOFs) represent an ideal playground in the search for new adaptive materials[27]. MOFs can host gas molecules and, due to their tailored porosity, enables control of matter and heat exchanges. This is a fundamental feature in many living systems; for example, plant leaves are capable of self-adjusting water content by stomatal pore opening or closure[28]. MOFs also offer key advantages in constructing composites and sophisticated devices; they can now be synthetized and shaped in a myriad of forms such as nanoparticles, films or patterns enabling their integration into optical devices[29–31]. However, programming autonomous responses in MOF materials requires a conceptual shift in how we conceive responsive porous-based materials. Unlike classical "stimuli-responsive" porous materials[32,33], self-regulating systems do not require any external control to switch states; rather, they adapt and evolve spontaneously under constant external perturbation. For instance, an optical self-regulating porous material would adapt its light absorption to counterbalance light irradiation, reminiscent of the functionality of the human eye[34]. More ambitiously, engineering such a process would allow the emergence of MOF-based materials programmed over time[24], for instance exhibiting spontaneous oscillations similar to those observed in living systems.

To fill this gap, we design a plasmonic metasurface coupled with MOFs that presents a thermo-optical self-regulating behavior. MOF hybrid plasmonic materials have been previously used for sensing, nanomedicine or catalytic applications as summarized in Supplementary Table 1. In this study, we introduce a previously unexplored and unique functionality of MOF-hybrid plasmonic materials that exploits the temperature-induced sorption behaviors of colloidal assembly of MOFs interconnected with plasmonic antennas to integrate an internal negative feedback mechanism and induce optical self-regulation. Interestingly, we show that this thermo-optical-sorptive process is dynamic: each individual MOF/antenna system exhibits self-

oscillatory behavior in presence of constant external energy input, working as a light-fueled nano-steam engine.

## Results

### Design of the MOF plasmonic metasurface integrating a negative feedback mechanism

The self-regulating device, illustrated in Fig. 1, is able to adapt its optical properties (optical absorptivity) to the incoming light intensity. Its design requires the integration of a thermo-optical negative feedback mechanism that we devised by coupling two materials characterized by crossed sensing/actuating responsiveness. First, a plasmonic metasurface made of an array of antennas[35,36], as illustrated in Fig. 1b, capable of optical sensing and efficient thermal heating in presence of a continuous flow of energy (laser at 850 nm in our case). The photoheating capability of each plasmonic antenna depends on its absorptivity, usually quantified through the wavelength-dependent absorption coefficient that defines the local fraction of incident light intensity that is absorbed by the material[37]. The heat generation (Q) in plasmonic antenna due to absorbed light can be described by:

$$Q = \alpha I_0 \tag{1}$$

where Q is the heat generated per unit volume, $\alpha$ is the absorptivity of the antennas and $I_0$ is the intensity of the incident light.

Second, a colloidal MOF film that "senses" the temperature by inducing an optical change. The process is mediated by condensation/evaporation of a liquid/vapor phase. In our case, the colloidal MOFs are constituted by ZIF-8 colloidal films in the presence of isopropyl alcohol (IPA) vapor[31]. The simple functioning of this thermo-optical layer is schematized in Fig. 1b: cooling down the film, the IPA condensates into the porosity. The porous film fulfilled with IPA is in the so-called "Full" (F) state; at low temperatures the refractive index of the ZIF-8 film ($n_{ZIF-8}$) is high. Conversely, when heating the film, desorption takes

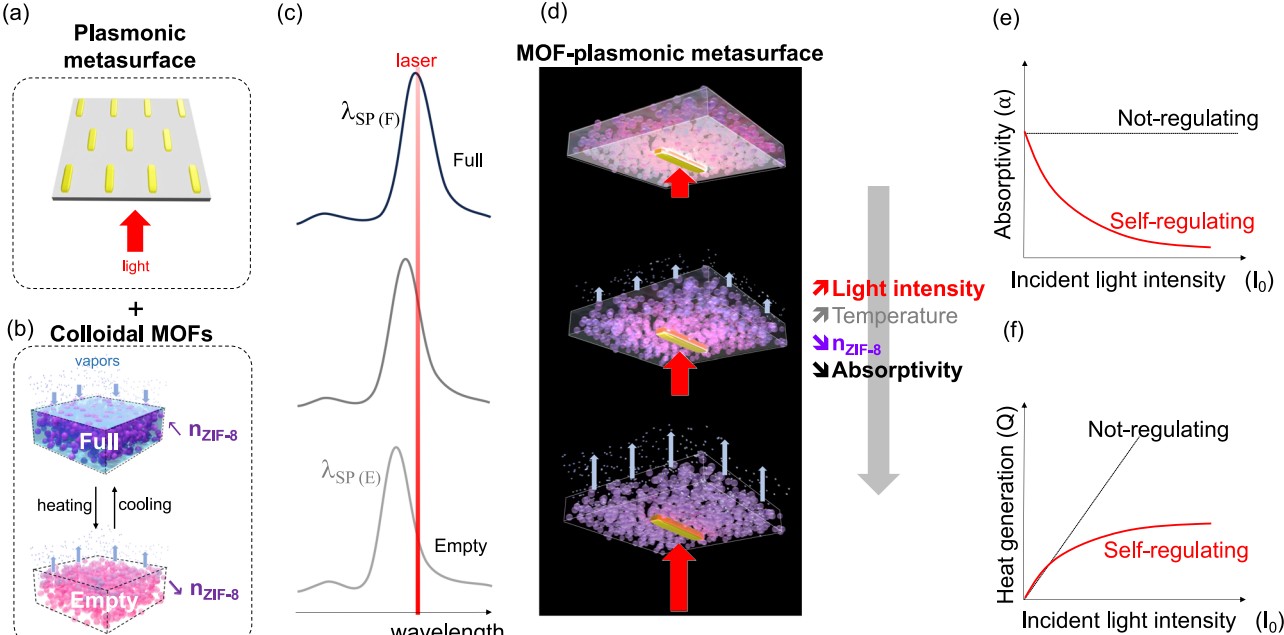

**Fig. 1 | Functioning of the self-regulating device.** Illustration of the **a** plasmonic metasurface made of an array of nanoantennas, **b** thermally driven optical response of a MOF colloidal layer in the presence of vapor. **c** Evolution of the absorption plasmonic curves and **d** corresponding illustration as a function of light intensity.

The gray arrow indicates the evolution of light intensity, temperature, $n_{ZIF-8}$ and absorptivity. Expected evolution of the **e** absorptivity and **f** heat generation as a function of light intensity for the self-regulating device compared to a conventional non-regulating material.

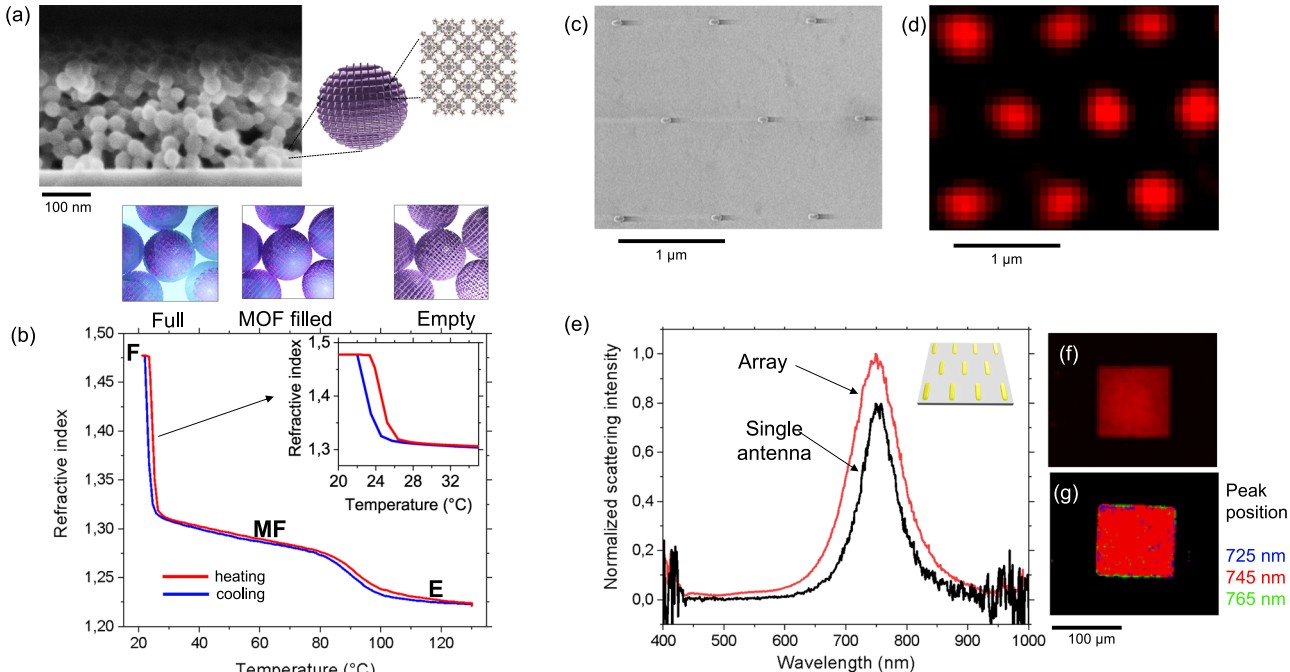

**Fig. 2 | Optical characterization of the MOF layer and of the metasurface. a** SEM micrograph of the ZIF-8 colloidal film. **b** Refractive index as function of the temperature of the ZIF-8 layer in presence of 0.7 P/P₀ of IPA. The inset highlights the hysteresis between the F and MF states. **c** SEM micrograph of the plasmonic antennas on glass. Hyperspectral micrographs in dark field mode of (**d**) the single antennas. **e** Scattering intensity as function of the wavelength of the full array and of the single antenna. The inset illustrates the plasmonic array. **f** Hyperspectral micrographs in dark field mode of full array and **g** color-map of the plasmonic scattering peak position of the 10,000 antennas of the array.

place leading to an "Empty" (E) state and a decrease in the refractive index of the ZIF-8 film.

The operation of the self-regulated layer is shown in Fig. 1c, d. The plasmonic antenna is coupled with the ZIF-8 layer; in presence of vapor, the system is in the "Full" state and the plasmonic antenna exhibit a longitudinal surface plasmon resonance corresponding to a peak at $\lambda_{SP\,(F)}$. When applying a laser at wavelength matching the $\lambda_{SP\,(F)}$, the system is able to adapt its response to the power of the incoming laser. At low laser power, the plasmonic absorption causes only a limited local heating of the particle and of the surrounding ZIF-8 layer. The system remains close to the "Full" state with a high absorptivity. Conversely, when increasing the laser intensity, the temperature rises and desorption occurs; this step is accompanied by a decrease in refractive index of the ZIF-8 ($n_{ZIF-8} \downarrow$); the plasmon absorption peak is gradually blue-shifted toward the $\lambda_{SP\,(E)}$, out of the "excitation window". This shift results in a decrease of absorptivity at the wavelength of the laser. Increasing the incoming light power, the MOF plasmonic metasurface will absorb less light, converting less energy into heat, and thus result in lower plasmonic heating efficiency. The composite metasurface ultimately behaves as a homeostat at the nanoscale capable of counterbalancing the effect of the incoming light reducing its absorbed intensity, Fig. 1e, as well as the resulting generated heat with respect to a non self-regulating antenna as shown in Fig. 1f.

**Fabrication of the device and in situ single-antenna analysis**
The fabrication of such self-regulating metasurface requires a very strict calibration and interconnection of its components. For the colloidal ZIF-8 film, the main requirement was a film with a very high porous volume to induce a large variation of refractive index (and thus $\Delta\lambda_{SP}$) as a function of the temperature. From a practical point of view, ZIF-8 colloids are convenient because they can be applied directly onto plasmonic materials at room temperature, forming films of optical quality and can easily be dissolved (with acetic acid) and re-applied if necessary. The SEM micrograph in Fig. 2a displays a cross-section view

of ZIF-8 colloidal film obtained after synthesis and deposition. The film is composed of nanoparticles having an average size of 40 nm and has a thickness of around 200 nm. The crystallinity of the material was first confirmed by X-ray diffraction as shown in Fig. S1. The thermo-optical response of the ZIF-8 colloidal film in the presence of IPA vapors was tested by in situ ellipsometric thermo-porosimetry[38]. Figure 2b illustrates the adsorption and desorption isobars carried out in presence of a constant relative vapor pressure of IPA = 0.7 P/P₀$^{(20\,°C)}$ and expressed as the evolution of refractive index (at 700 nm) as a function of the temperature of the film. At $T$ = 120 °C, the colloidal film presents a low refractive index of 1.22 suggesting that the film is "empty" (E). Decreasing the temperature, we observed an increase in refractive index around 90 °C corresponding to IPA filling of the ZIF-8 microporosity. At 60 °C the porosity of the MOF itself is filled (state called "MF") but not the inter-particle voids. Further cooling leads to the capillary condensation of IPA molecules in the mesopores formed by inter-particle voids. At room temperature (20 °C) the whole film is in the Full state (F) corresponding to a refractive index of 1.475 By heating back, the process is reversible. The films present a hierarchical porosity (interparticle + intraparticle) ensuring a porous volume of 63% inducing a refractive index variation of around 0.25 between the (F) and the (E) states as detailed in Fig. S2 and in agreement with previous reports[31]. Importantly, the sorption transition between the F and MF states presents a hysteresis as displayed in the inset of Fig. 2b.

For the design of the plasmonic metasurface, we choose anisotropic Au antennas because of their narrow plasmon resonance, their refractive index sensitivity and their photothermal capability[39]. The size of the antenna was optimized to align $\lambda_{SP}$ aligned with our light source (laser at 850 nm) in the "full porosity" (F) configuration. We designed the antenna by optical simulation starting from the value of refractive index of ZIF-8 in Fig. 2b as detailed in Figs. S3 and S4.

The antennas were made by electron beam lithography by preparing arrays with different sizes as detailed in Fig. S5. The optimized Au nanoantenna array is a squared pattern of 100 μm × 100 μm on

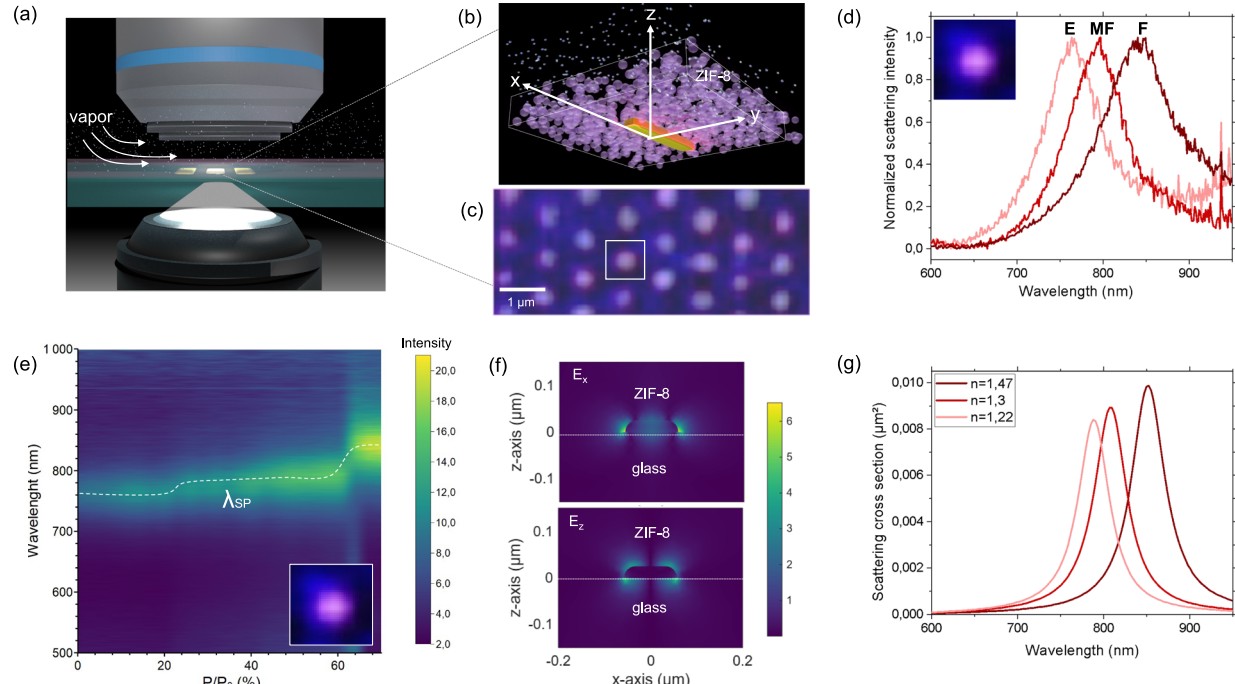

**Fig. 3 | Sorption behavior at the single antenna level. a** Illustration of the in situ environmental hyperspectral microscope. **b** Illustration and **c** hyperspectral scattering micrograph of the plasmonic/MOF composite. Adsorption isotherm at the single-antenna level: **d** Experimental single-antenna scattering spectra corresponding to the empty (E), MOF-filled (MF) and full (F) states; **e** evolution of the scattering spectra as function of the P/P$_{O\ (IPA)}$; the dotted line is a guide for the eye. **f** Simulated electric filed distribution around a single antenna along the symmetry plane parallel to its long 120 nm axis. The simulation indices are indicated in the inset. **g** Simulated single-antenna scattering spectra corresponding to the empty (E), MOF-filled (MF) and full (F) states.

glass. Figure 2c displays a SEM image of 120 nm long antennas (width 30 nm and 20 nm height). They are organized in a periodic array with a periodicity of 1 μm (surface density 1 per μm²), a distance sufficient to make behave the antennas as optically and electronically isolated units. To probe the optical response (scattering) of the full array and of each individual antenna, we performed hyperspectral microscopy in dark-field mode[40–42] as shown in the micrographs in Fig. 2d. The scattering spectra of the full array and that of an individual antenna, plotted in Fig. 2e, both shows a plasmonic peak around 745 nm. By analyzing the hyperspectral micrograph of the full array shown in Fig. 2f we could map the position of the peak position over the full array of 10,000 antennas. The color-map in Fig. 2g indicate the position of the plasmonic scattering across the whole array; 97% of the sample consists of plasmonic antenna with peak centered at 745 ± 10 nm, confirming the spectral "homogeneity" of the plasmonic array from a spectral viewpoint.

The plasmonic metasurface on glass has been covered by spin-coating deposition with a 200 nm ZIF-8 colloidal layer, as determined by ellipsometry. The optical response of each individual antenna in presence of IPA vapors was probed by in situ hyperspectral microscopy customized with an environmental chamber in dark-field mode as shown in Fig. 3a, b. The hyperspectral micrograph in Fig. 3c displays the scattering signal of several individual plasmonic antenna covered by the ZIF-8 layer. During the experiment, the relative vapor pressure was progressively increased from 0 to 0.75 while recoding the scattering spectra of each individual antenna. The evolution of the scattering intensity during adsorption of one individual antenna as function of the wavelength is reported in a 2D plot in Fig. 3e. This adsorption isotherm at the single antenna level is highlighted with the dotted line. The curve exabits two main red-shifts during vapor sorption: (i) from 788 to 805 nm due to IPA uptake into the microporosity of the ZIF-8 colloids (E→MF) and (ii) from 805 to 850 nm due to IPA uptake in the interparticle porosity (MF→F). The scattering curves

corresponding to the three states (E), (MF) and (F) are displayed in Fig. 3d. As expected, the plasmonic antennas are sensitive to the refractive index of the surrounding ZIF-8 in the vicinity of the Au particles. This sensitivity is higher in the vicinity of the tip of the anisotropic antenna as confirmed by the local enhancement of electric field obtained by simulation of Fig. 3f. In addition, the simulation clearly indicates that, antenna spaced 1000 nm do not exhibit coupling effect and behave as optically individual antennas. We then calculated the scattering spectra starting from the refractive index of the ZIF-8 layer in Fig. 2b. The simulated spectra shown in Fig. 3g are in good agreement with the experimental results; these first results confirm that the scattering curve for this antenna in the "F" state is centered around 850 nm, the wavelength of our laser source, meeting the main requirement for obtaining the self-regulating device.

## Self-regulating metasurface

We then tested the capabilities of the MOF plasmonic metasurface to modulate its adsorption when exposed to monochromatic light. To do so, we used a microscope equipped with a NIR-sensitive camera and with the environmental chamber as described above. In addition, we exposed the surface to an 850 nm laser through the dark-field condenser in order to simultaneously irradiate and image the scattering evolution of the sample (Fig. 4a). More specifically, we imaged and quantified the scattering intensity at 850 nm (I$_{850nm}$) of the array as a function of the incoming laser intensity(I$_{laser}$)[42]. The intensity-based plasmonic signal, was determined by following the evolution of the scattering intensity normalized by the incoming laser intensity. As plotted in Fig. S6, following the evolution of antenna's scattering intensity provides a direct insight on the evolution of the absorptivity as demonstrated by simulations in Fig. S7. To validate the concept illustrated in Fig. 1, two experiments were conducted, in presence of IPA vapor and without vapors as shown in Fig. S6. In absence of vapor, the normalized I$_{850nm}$/I$_{laser}$ ratio remains constant upon increasing the

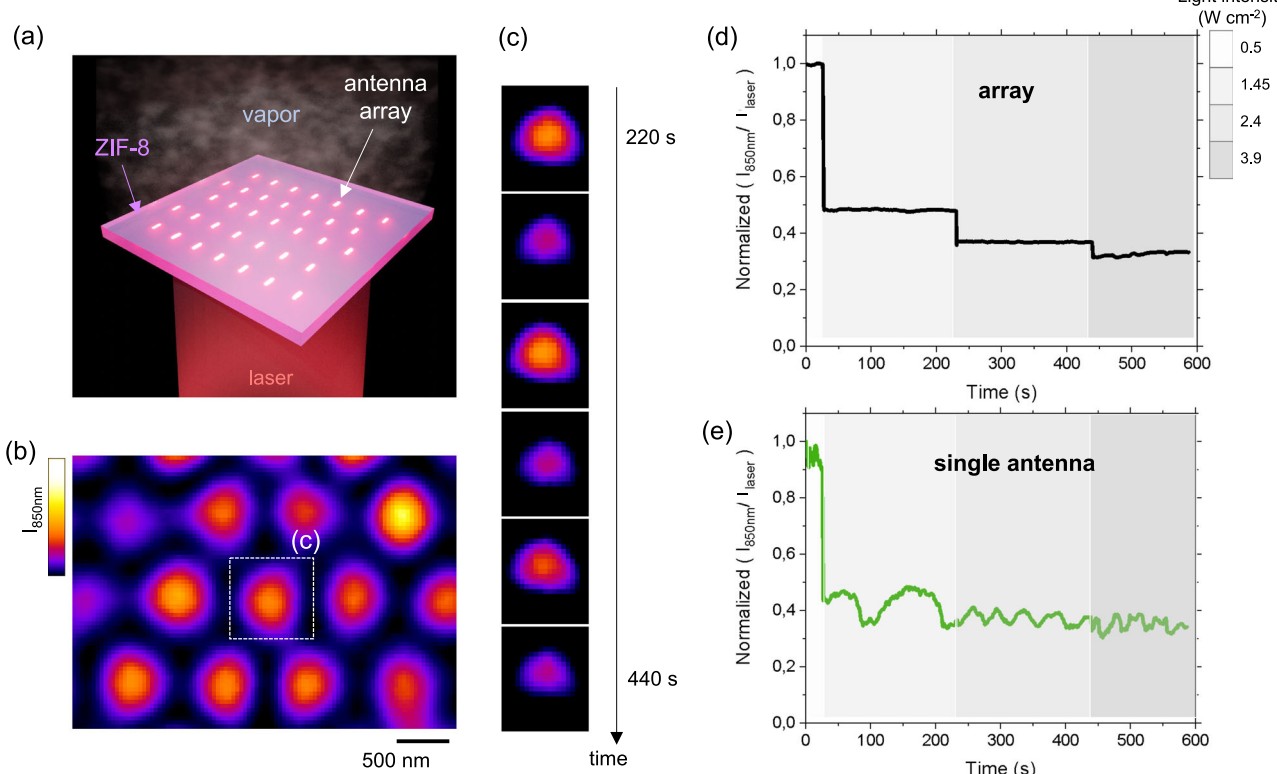

**Fig. 4 | Self-regulating metasurface. a** Illustration of the MOF plasmonic nano "steam engines" array. **b** Micrograph showing the scattering at 850 nm of each individual antenna and **c** time sequence of the evolution of the scattering of a single antenna irradiated with a laser intensity of 2.4 W/cm². The gradient color scale represents the variation in normalized scattering intensity, expressed in arbitrary units. **d** Evolution of the normalized scattering intensity over time for the entire array. **e** Evolution of the normalized scattering intensity over time at the single antenna level.

laser intensity. Indeed, in this configuration the material is in the (E) state, it is not self-regulating and the absorptivity α remains constant. Instead, when adding IPA vapor with a vapor pressure of 0.7 $P/P_0$, the material is initially in the (F) state with a plasmonic peak centered around 850 nm. In this configuration, the normalized $I_{850nm}/I_{laser}$ ratio decreases when increasing the laser power because of the shift of the plasmonic peak induced by the heating confirming the self-regulation properties as envisioned in Fig. 1. More precisely, Fig. S6 indicates that the MOF plasmonic metasurface is able reduce the scattering and absorptivity by 60% when the laser intensity is increased from 0.05 to 3.9 W cm⁻². Based on these data, we can estimate the extent of desorption and the local temperature as a function of the incoming laser power (Fig. S8). The analysis shows that increasing the laser intensity from 0.05 to 3.9 W cm⁻² leads to an estimated temperature rise of about 6.8 °C. Under our irradiation conditions (up to a maximum 3.9 W cm⁻²), the system transitions from the (F) state to the (MF) state. Higher temperatures required to reach the fully empty (E) state (up to ~110 °C) are not achieved under our experimental conditions, due to both the limited laser power and the self-regulating nature of the device, which reduces light absorption at higher intensities. The self-regulating response of the system is not uniform across the entire temperature range, but instead exhibits distinct regions of enhanced sensitivity. This behavior can be analyzed by calculating the derivative of the scattering signal at 850 nm with respect to temperature. As shown in Fig. S9, the highest sensitivity occurs between 22 °C and 27 °C, corresponding to the transition from the Full (F) to the MOF-filled (MF) state. A second, less intense sensitivity peak is observed between 80 °C and 100 °C, associated with the transition from the MF to the E state. These findings suggest that the self-regulating effect is most efficient within the 23–27 °C window. Importantly, the sensitivity profile is strongly influenced by the adsorption/desorption dynamics

within the porous layer. As a result, the system's operating range can be readily tuned by altering the porosity characteristics, the chemical nature of the vapor, or the vapor concentration ($P/P_0$), offering flexibility for application-specific optimization. We explored the possible effect of the vapor pressure. As demonstrated by in situ ellipsometry experiments conducted at different vapor pressures (0.5, 0.7, and 0.9), vapor pressure systematically shift the sorption equilibrium (Fig. S10). Specifically, higher vapor pressures delay desorption, thereby shifting the thermo-optical response of the MOF layer toward higher temperatures, while lower vapor pressures produce the opposite effect. This behavior indicates that maintaining a controlled vapor pressure is essential for reliable device operation. At the same time, this variability presents a significant opportunity: by adjusting the vapor pressure, one can, in principle, dynamically tune the temperature range over which the device exhibits its large self-regulating response.

### Self-oscillatory behavior: nano "steam-engines"

The next step was to investigate the dynamics of this self-regulation process. To do so, we examined the evolution of plasmonic scattering over time for both the array, Fig. 4a, and at the single antenna level. Specifically, Fig. 4b shows a micrograph of the scattering signal (at 850 nm) from the plasmonic antennas covered by the ZIF-8 layer. We first analyzed the dynamic response of the full array, Fig. 4d. Consistent with Fig. S6, increasing the laser intensity leads to a decrease in normalized scattering. When the laser power is kept constant, the scattering value remains steady, showing no notable dynamic response. In reality this seemingly steady response results from the averaged dynamic contributions of each individual antenna as shown in Video 1. When examining the temporal response of a single antenna, we observed, in some cases, marked oscillatory signals in the scattering intensity in presence of constant light input, as shown in Fig. 4c for

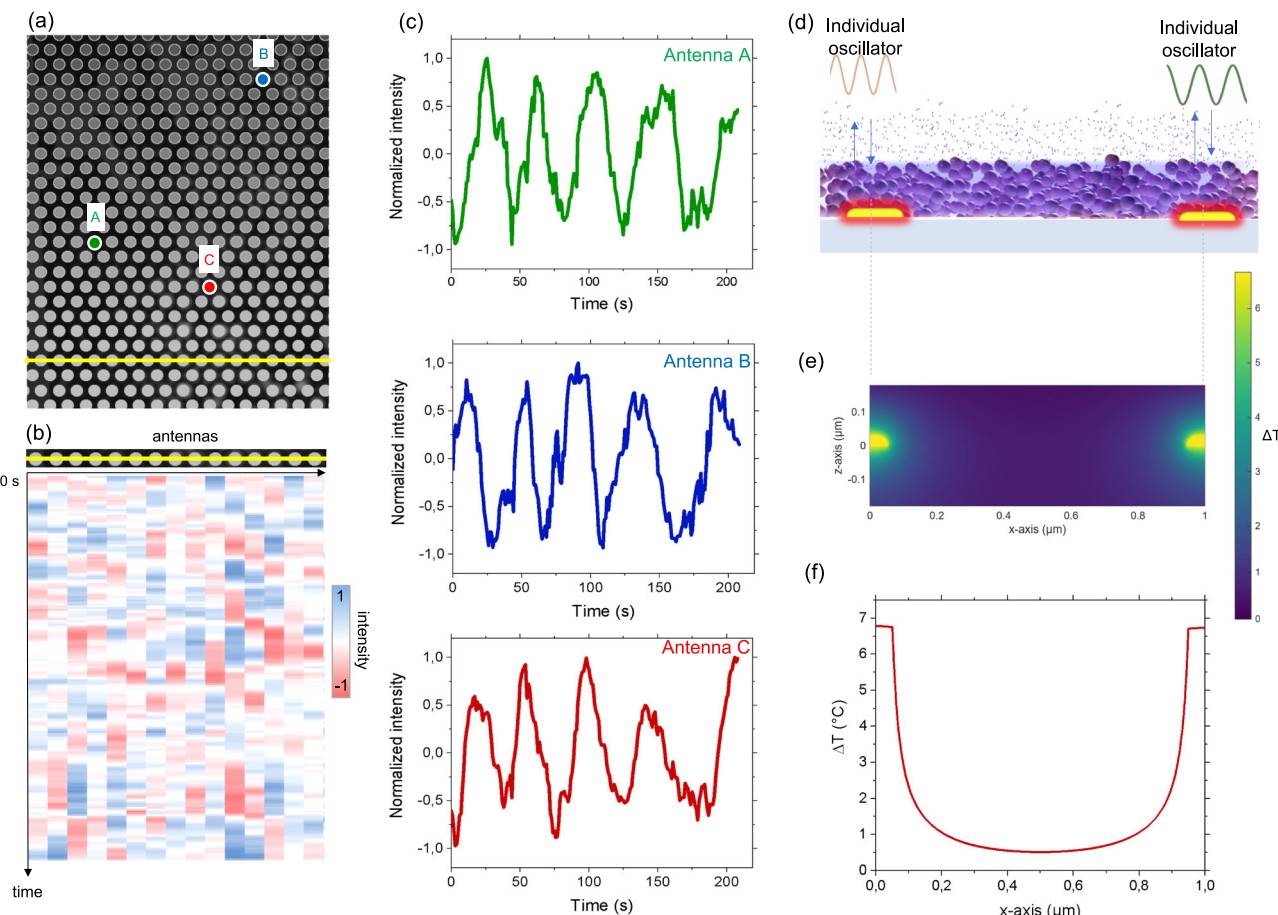

**Fig. 5 | Dynamic response at the single antenna level. a** Scheme of the ZIF-8/antenna array displaying the selected antennas. **b** Temporal evolution of the normalized scattering intensity of a row of ZIF-8/antennas. **c** The similar oscillatory evolution of the three representative ZIF-8/antennas not placed close to each other. **d** Illustration and **e** simulated temperature distribution of two neighboring ZIF-8/antennas. **f** Simulated temperature rise as a function of the distance along the x-axis at the glass surface.

a representative antenna. The time evolution of normalized scattering of a typical single antenna under different light irradiations is displayed in Fig. 4e.

For each laser intensity, the scattering intensity periodically fluctuates around an equilibrium state. To rule out the possibility that the observed oscillatory behavior was due to artifacts, we challenged our experiment by conducting it without vapor and without the 850 nm laser (by using a white light instead). In both cases, no oscillations and no collective fluctuations were observed (this aspect will be further discussed later on).

We then conducted a detailed image analysis to reveal the dynamic behaviors of each individual ZIF-8/antenna on the array for different irradiation conditions. To do so, we developed an automatic procedure to track and analyze the dynamic evolution of each antenna. As discussed in SI, we ranked the ZIF-8/antennas based on the frequency and "quality" of their oscillations, with periodic oscillations considered of high quality. For clarity, we mainly discuss here with the case of a laser intensity of 2.4 W/cm². Considering the full array (Fig. 5a and Fig. S11), a variety of behaviors were observed. As seen in Video 1, while most of the antennas oscillate, no collective synchronous behavior is observed. At first glance, some antennas oscillate periodically, others oscillate only slightly or randomly not necessary depending on their locations. In addition, different antennas oscillate at different frequencies as shown in Figs. S12, 13 and 14. For instance, the similar oscillatory behavior of three ZIF-8/antennas, named A, B and C is shown in Fig. 5c. The three antennas, which are not located close to each other, exhibit a similar frequency but are not at all in phase (Fig. S15). We then investigated the

oscillatory behavior of neighboring antennas that are close to each other. Figure 5b shows the evolution of the normalized scattering intensity at 850 nm as a function of time for a series of ZIF-8/antennas located on the same row. The 2D plot demonstrates that the antennas exhibit spontaneous fluctuations in scattering intensity. However, these fluctuations are neither in phase nor collective.

To further investigate the possible origin of these different oscillatory behavior, we performed an in-depth spectroscopic mapping at the single antenna level of the metasurface covered with ZIF-8 and in the presence of IPA vapor. While bare antennas exhibit homogeneous optical behavior (Fig. 2), the presence of ZIF-8 and vapor induces a certain variation in the scattering response of neighboring antennas (Fig. S16). The ZIF-8 film consists of randomly stacked colloidal particles. Given that the antennas are particularly sensitive to refractive index changes at their extremities (as shown in Fig. 3), the non-uniform distribution of ZIF-8 and condensed vapor results in variations in absorptivity (α) at 850 nm and different local heating that is responsible for the oscillatory process. This unintended variability can be seen as an advantage to support our approach as it provides an additional evidence that the oscillations are not due to artifacts. For instance, possible fluctuations in vapor pressure or in incident light intensity would result in a simultaneous variation of the scattering intensity for all the antennas. Instead, these results indicate that each individual ZIF-8/antenna composite behaves as an individual oscillator, as illustrated in Fig. 5d. These observations also confirm that by controlling the optical properties and the porous medium, different oscillatory behaviors can be obtained.

The different oscillatory behaviors also suggests that thermo-optical process occurs at the single antenna level with very low thermal interaction with neighboring antennas. To support this hypothesis, we performed thermal simulations of two antennas acting as local photo-heaters under continuous light irradiation (Fig. 5e). As an example, we simulated a local temperature increase of 6.8 °C, in agreement with the value determined above. Figure 5f shows the temperature profile at the glass surface ($z = 0$) along the x-axis; the temperature rise is located in the close vicinity of the antenna with a very low heating between the antenna.

The possible mechanism at the heart of this "time-programmed" behavior is analogous to a nanometric steam engine and is illustrated conceptually in Fig. 6a. We can describe the self-regulating system with a "seesaw" analogy, where the two extreme cases are represented by the (F) and (E) states, characterized by the plasmonic curves. When the laser is applied, local heating results in a new equilibrium position (green curves) characterized by lower absorptivity. In this configuration, the system is not steady but dynamically self-regulates between two intermediate states (green curves) by alternating partial adsorption and desorption of IPA into/from the ZIF-8 layer (Fig. 6c). This periodic process results in shifts of the plasmonic band peak between the two states and consequently, in oscillations in the scattering intensity as shown in Fig. 6b.

The oscillatory behavior is due to the presence of a sorption hysteresis between the F and MF states (Fig. 2b). In addition, there is a time delay between the fast plasmonic heating (microseconds) and the slower adsorption/desorption response (seconds), with time constants that we have experimentally determined, as discussed in Figs. S19 and S20.

To explore a possible mechanism underlying the self-oscillatory behavior, we employed simulation based on a simple hysteresis-based model grounded in the experimentally determined kinetic parameters and hysteresis behavior of the MOF layer. This model allows us to explore the influence of various parameters, in particular the shape of the hysteresis curve and laser intensity on the oscillation dynamics (SI). As shown in Fig.6d, we modeled our system using a hysteresis curve in which the absorptivity (at 850 nm) decreases with temperature, consistent with the results in Fig.2b and Fig. S8c. As shown in Fig. 6e, the analysis displays an oscillatory behavior in absorptivity with a period of around 45 seconds, which is comparable to our experimental observations. The system demonstrates absorptivity oscillations that results from an elliptical loop in the hysteresis curve, capturing the delayed transitions between heating and cooling states that drive the self-oscillatory behavior. The shape and path of this hysteretic loop has an thus impact the frequency of the oscillation. To further explore that, we performed the same analysis using hysteresis curves with identical threshold temperatures but varying steepness parameters ($\beta$), Fig. 6f, which control how sharply the adsorption and desorption transitions occur (as defined in Supporting Information Fig. S21). This variation can be describe porous networks with different pore size distribution: monodisperse pores lead to sharper transitions, while a polydisperse distribution results in broader, less abrupt changes[43]. As shown in Fig. 6g, the frequency of the oscillation increasing for shaper hysteresis. Similarly, as shown in Fig. S22, broader hysteresis results in a larger loop and a decrease of the frequency (no oscillation are observed for no hysteresis or very narrow hysteresis). This simple analysis suggests that by varying the sorption properties and optical response of the porous media, different oscillatory behaviors can be achieved. This may further explain why heterogeneous packing of ZIF-8 on the antenna leads to variations of photo-heating and sorption profile and thus oscillatory behavior as discussed in Fig. 5.

Finally, the effect of light intensity was examined. Observations on selected antennas (Fig. 6b), suggest that increasing the light intensity may induce a higher oscillation frequency. We performed simulations with increasing light intensity, as shown in Fig. S23. Interestingly, we confirmed that the oscillation frequency increases with light intensity until the absorptivity reaches a low value, where the steepness of the hysteresis decreases, causing the oscillations to slow down and then stop.

## Discussion

The self-regulating performance of our metasurface can be compared to other self-regulating systems reported in the literature. Focusing specifically on devices that are based on thermo-optical negative feedback, such as a homeostatic photonic device[44], liquid crystal elastomers[19], or materials based on phase transitions[45] like $VO_2$[46], all these systems are designed to respond to temperature increases with an opposite optical response as summarized in Fig. S24. The mechanisms of feedback; the functioning conditions and the type of optical response absorptivity, emissivity, transmittance) differ among these approaches and each approach comes with its own advantages and limitations. However, one useful comparison metric is the thermo-optical sensitivity, defined as the ratio between the relative change in the optical property and the corresponding temperature range over which the change occurs. In general one of the main challenge is have this value the highest possible in the temperaure range of interest. In this context, even though a direct comparison is not straightforward, our MOF plasmonic metasurface exhibits a significantly higher sensitivity to temperature changes in the range close to ambient temperature (60% decrease in absorptivity heating from 20 to 26.8 °C).

Concerning the self-oscillating behavior, each individual ZIF-8/antenna composite behaves as an individual self-regulating and self-oscillating device fueled by a constant external energy input, resembling a thermo-optical nano-steam-engine as conceptualized in Video 2. In our experiments, each oscillator evolves independently. In perspective, one possibility would be to induce thermal coupling by bringing antennas closer together as indicated by the simulation in Fig. S18; by doing that, it may become possible to heat even more efficiently, fine-tune the oscillatory dynamics and achieve deliberate synchronization across groups of antennas[47]. In terms of programming capabilities, the implementation of a dynamic vapor-mediated thermo-optical feedback mechanism, open possibilities for the control on the oscillatory behavior. As suggested by simulation, more refined control over these oscillations could be achieved by exploring different porous materials (different hysteresis), by tuning light intensity, by modifying the vapor environment or the design of the plasmonic antennas. This level of control could unlock new functionalities rhythm-based materials capable of sending signals and transfer information for diverse applications in robotics, optical manipulation[48], sensing, information processing[49] or even computing where precise time-dependent behaviors are essential.

Summarizing, we engineered MOFs to fabricate a hybrid metasurface capable of autonomously adapting light absorption as a function of the incident light irradiation. To achieve this, we coupled colloidal MOFs with plasmonic antennas in order to design a thermo-optical negative feedback mechanism. We harnessed temperature-induced vapor sorption in/from the colloidal MOFs and the photo-thermal effect of the plasmonic antennas as a source of heat. More specifically, we demonstrated that the MOF-plasmonic metasurface autonomously adapts its optical properties to changes in incoming light intensity by reducing its absorptivity and scattering by up to 60% as the light intensity increases. Interestingly, we observed that this self-regulated process is dynamic and predominantly localized at the nanoscale: the individual MOF/antenna of the array exhibits self-oscillatory fluctuations due to vapor reminiscent of the functioning of a steam engine at the nanoscale.

Beyond typical stimuli-responsive devices, this work showcases the great potential of engineering MOF and sorption processes to fabricate "more autonomous" optical metasurfaces for applications in adaptive optics, smart windows for thermoregulation or sensing. In

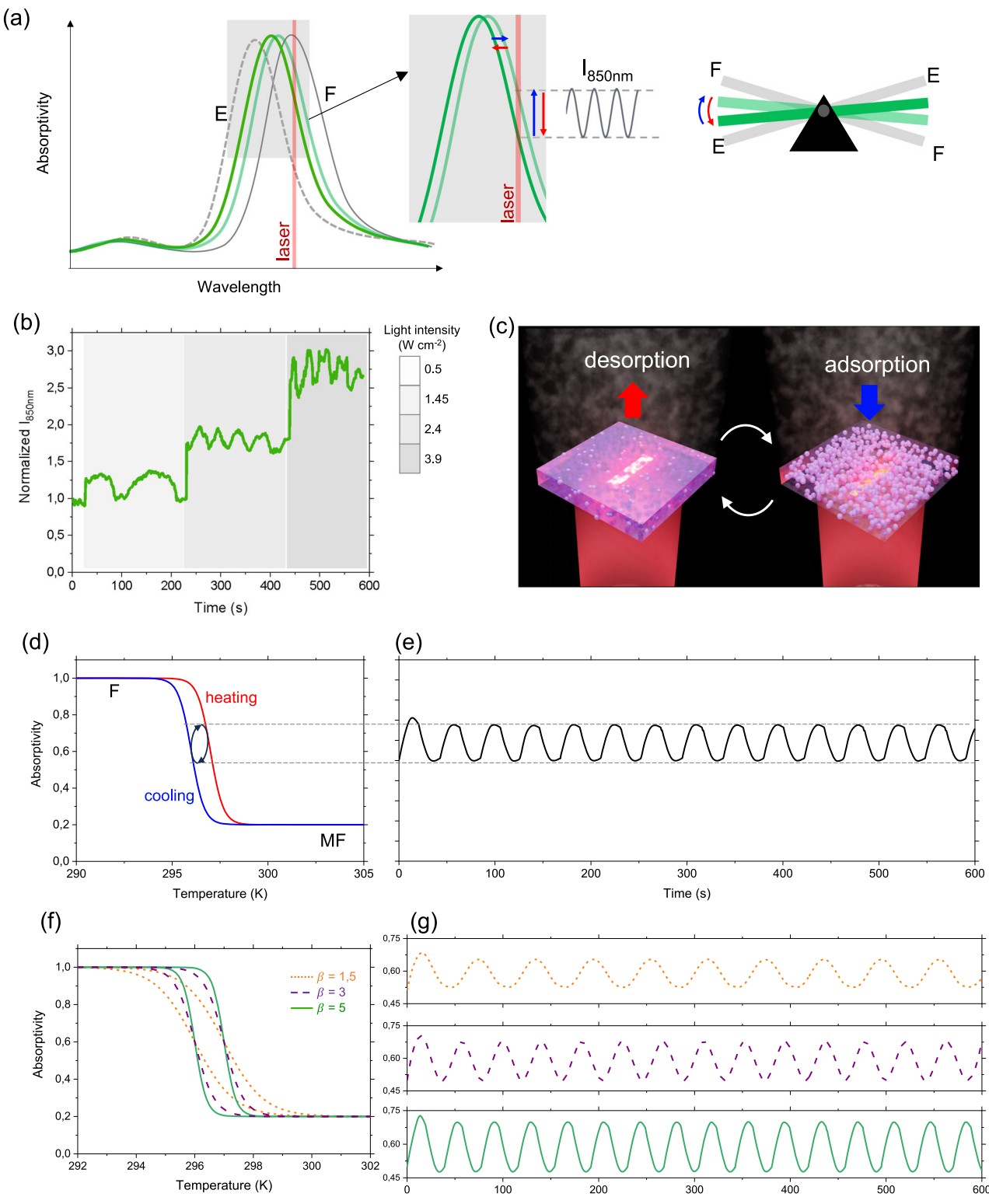

**Fig. 6 | Self-oscillating nano "steam engines". a** Mechanism of self-oscillations: absorptivity curves illustrating the (F) and (E) states (gray curves) and the two intermediate states (green curves); 'seesaw' analogy and zoom of the two plasmonic curves representing the two intermediate states, between which the oscillations occur, resulting in an oscillation of scattering intensity at 850 nm. **b** Non-normalized scattering intensity over time at the single antenna level. **c** Illustration of the self-regulating mechanism at the single antenna level. **d** Hysteresis curve of absorptivity (at 850 nm) as fonction of the temperature (red and blue curves correspond to heating and cooling) and **e** corresponding simulated oscillatory evolution. **f** Hysteresis curve of absorptivity (at 850 nm) as fonction of the temperature with different steepness paramenters and **g** corresponding simulated oscillatory curves.

addition, given the approach's versatility, it holds significant potential for further exploration with other porous materials, optical absorbers as well as various vapor molecules, enabling enhanced control over the global self-regulating response and local oscillatory dynamics, i.e., in amplitude and frequency. Looking further ahead, the observed oscillatory response at the nanoscale marks a first step for the developement of dynamic porous nanomaterials with integrated clocks or rhythms and with life-like autonomy.

## Methods

### Fabrication of the colloidal ZIF-8 films

The colloidal ZIF-8 solution was synthesized following the procedure detailed in the literature, with slight modifications to the protocol originally reported by Cravillon et al.[50]. To prepare the solution, 1.47 g (4.94 mmol) of $Zn(NO_3)_2 \cdot 6H_2O$ was dissolved in 100 mL of methanol. This solution was then mixed with 3.24 g (39.46 mmol) of 2-Methylimidazole, which had been previously diluted in 100 mL of methanol. The mixture was allowed to cool and was gently stirred. Within seconds, the solution turned turbid. After 7 min, the mixture was centrifuged multiple times to separate the nanoparticles from the dispersion. The resulting nanoparticles were then diluted with methanol and spin-coated onto the antenna sample.

### Fabrication of the Au plasmonic array

The fabrication of plasmonic metasurface was carried out using electron beam lithography (EBL). Initially, the quartz substrate underwent nitrogen plasma treatment to clean the surface and enhance adhesion. Following this, a copolymer layer of EL 12 (MMA(8.5)MAA) was spin-coated with a thickness of 55 nm, and then pre-baked at 160 °C for 10 min. Subsequently, a layer of PMMA A2 was spin-coated to a thickness of approximately 50 nm and also pre-baked at 160 °C for 10 min.

To facilitate the (EBL) process, a 20 nm layer of aluminum was deposited via e-beam evaporation. The substrate was then exposed to a 100 KeV electron beam. Post-exposure, the aluminum layer was removed using a 0.25 M NaOH solution. The development of the PMMA/MMA layers was achieved by immersing the substrate in a solution of Methyl isobutyl ketone and IPA (1:3) for 30 s, followed by rinsing in IPA for another 30 s.

For the final metal deposition, a thin layer of 1 nm germanium and 20 nm gold was evaporated using e-beam evaporation. The lift-off process was performed in an SVC-14 stripper at 80 °C to remove the remaining resist and leave behind the patterned nanoantennas. A final oxygen plasma treatment was conducted to remove any residual resist.

### Characterization methods

Scanning electron microscopy (SEM) imaging was performed with SU-70 Hitachi FESEM, equipped with a Schottky electron emission gun[51]. Powder X-Ray diffraction was carried out using a D8 Advance diffractometer (Bruker), equipped with a Cu anode (K$\alpha$1 = 1.54056 Å and K$\alpha$2 = 1.54439 Å) and a 200 channels LynxEye detector.

### In situ environmental spectroscopic thermo-ellipsometry

The ZIF-8 films were characterized using a UV-visible-NIR Woollam spectroscopic ellipsometer (SE) operating from 240 to 1700 nm. This ellipsometer features a programmable heating stage (ranging from −80 °C to 600 °C) and an environmentally controlled chamber, allowing precise control of the films' local environment in terms of vapor pressure[52]. Data measurement and analysis were conducted using CompleteEASE software, which involved fitting the raw data curves with a Cauchy model for non-absorbing materials. A mass flow controller (SOLGELWAY) managed the vapor $P/P_0$ by combining two gas flows: (i) dry air flow with solvent $P/P_0 = 0$, and (ii) air flow that passed through a bubbler containing liquid isopropyl alcohol (IPA) $P/P_0 = 1$.

### In situ hyperspectral microscopy

The spectral imaging was performed by a Cytoviva hyperspectral microscopy (HIS) system in dark-field mode. The scattered light from the nanoantennas was collected using a 10 or 100x objective with an exposure time of 0.25 s.

The resulting hyperspectral image was analyzed using ENVI software, extracting the scattering spectrum for each pixel.

### Single antenna porosimetry

To performed the single antenna porosimetry, we integrated a closed environmental chamber at the HIS equipped with a gas flow and humidity controller (SOLGELWAY) and a temperature-controlled platform (Linkam Scientific)[53,54]. Hyperspectral analysis was acquired with an acquisition time of 0.250 s at each vapor pressure that we varied between 0 and 70 °C mantaining IPA partial pressure $P/P_0$ of 0.7.

### Light-driven self-regulation and self-oscillations

to investigate the self-regulation capabilities we employed the dark-field microscope with a continuous 850 nm laser (Changchun New Industries Optoelectronics Tech. Co.) in place of a white light source[55]. We used the 10x and 50x objectives to analyze the response of the full array and at the single antenna level respectively. Before every experiment, we first conducted measurements at lower laser power levels (with negligible heating) to establish the baseline and we kept all parameters constant (relative pressure at 0.7), to verify the absence of external fluctuations. Then, we systematically increased the laser power, recording the images and videos that stem from the changes in scattered laser intensity.

## Data availability

Data analysis from the main text or the Supplementary Information are provided with this paper. Source data are provided with this paper.

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

## Acknowledgements

This work was supported by the European Research Council (ERC) under European Union's Horizon 2020 Programme (Grant Agreement no. 803220, TEMPORE). We thank D. Montero and the Institut des Matériaux de Paris Centre (IMPC FR2482) for servicing FEGSEM & EDX instrumentation and Sorbonne Université, CNRS and C'Nano projects of the Région Ile-de-France for funding.

## Author contributions

H.A. synthesized the MOF films. H.A. and M.B. performed the thermal and optical simulations and provided guidelines to design the metasurfaces. H.B. validated the principles of thermal and optical simulations. A.C. designed and fabricated the Au metasurfaces. H.A and M.F. performed the ellipsometric and hyperspectral characterizations. H.A., D.R.C., and M.F. performed the light-driven dynamic experiments. D.R.C. developed the image analysis code. H.A., D.R.C., and M.F. developed the dynamic model and analyzed the data. H.A and M.F. wrote the

manuscript. All the authors contributed to the data interpretation, read and gave constructive inputs to the manuscript. M.F. conceived the idea and supervised the project.

## Competing interests

The authors declare no competing interests.
