## [Transparent Peer Review file · Nature Communications]

Self-regulating and self-oscillating Metal-Organic Framework hybrid plasmonic metasurfaces

Corresponding Author: Professor Marco Faustini

Version 0:

Reviewer comments:

Reviewer #1

(Remarks to the Author)

In this manuscript, the authors present an interesting work that integrates Metal-Organic Frameworks (MOFs) with plasmonic metasurfaces to achieve autonomous optical self-regulation. The observed self-oscillatory behavior at the single-antenna level is particularly noteworthy. However, several issues need to be addressed before the manuscript can be considered for publication.

1. The introduction lacks sufficient discussion of prior research in related fields, particularly in self-adaptation and self-regulation systems. Key information, such as evaluation metrics for self-regulation systems and existing challenges in the research field, is absent. A more thorough comparison with relevant literature would highlight the paper's novelty and provide clearer insights into the current state of research.
2. Like the negative feedback mechanisms of living systems, the dynamic range in the proposed system have not been explored. For instance, what is the minimum temperature change (sensitivity) that the system can respond to? Additionally, if the temperature exceeds a certain threshold (e.g. maximum laser power), does the system continue to function effectively?
3. The self-oscillatory behavior and optical self-regulation described in the paper appear to rely on a stable external energy input. However, the long-term stability and sustainability of the system under continuous operation are not discussed. Furthermore, the potential impact of environmental factors, such as temperature fluctuations and humidity changes, on system performance remains unexplored.
4. As discussed in Figure 5, "different antennas exhibit a similar frequency but are not at all in phase", which is somewhat unclear, given that the external stimulus (laser with the same intensity) should be the uniform across all units. Could this discrepancy be attributed to fabrication errors, such as the random distribution of the ZIF-8 colloidal film? A detailed discussion of the phenomenon is necessary, which is crucial for achieving better control over the self-oscillatory fluctuations.
5. More specific applications can be discussed, for instance, optical manipulation (Appl. Phys. Rev. 9, 031303 (2022)), information processing (Nature Communications 15 (1), 9658, (2024)), and others.
6. It's recommended that the authors pay more attention to enhancing the quality of the figures. The text is currently too small and difficult to read, and the overall aesthetics of the figures could be improved. These issues should be addressed to enhance the readability and clarity of the graphical content.

Reviewer #2

(Remarks to the Author)

In this manuscript, the authors report pioneering research on self-regulating adsorption and desorption of isopropanol and unique oscillation phenomena by combining adsorption/desorption characteristics in porous MOF thin films with local heating by plasmonic antennas. The oscillation phenomenon that occurs due to the coupling of changes in the plasmon resonance frequency caused by changes in the dielectric constant accompanying the adsorption and desorption of molecules (isopropanol) exhibit high novelty and is expected to be developed for various applications in the future. Since an important and highly novel discovery has been reported, it is appropriate to accept the paper after addressing the comments below.

- The introduction contains almost no background information on the MOF-Plasmonic array system used in this study. It is requested to add that the information on previous research using the MOF(porous materials)-plasmonic antenna system and the circumstances that led to the concept of this study.

- It is reported in the manuscript that the temperature change due to plasmonic heating is 6.3 degree C. The temperature change seems to be very small. It is necessary to show that the temperature dependence of the transition between the F-MF-E states of MOF (not just the relative temperature change) occurs under the temperature change estimated for the plasmonic heating in the experimental environment.
- A quantitative description of the oscillation frequency of plasmon scattering is required (around 50 s⁻¹), and additional information is needed on the factors that determine the oscillation frequency and the possibility of modulation.

Reviewer #3

(Remarks to the Author)

The paper by Amyar and coworkers deals with developing a self-oscillatory system based on thermoplasmonics and MOF components. The authors have proposed a conceptually fresh framework with the potential for fundamental studies of non-linear nanosystems and for developing nanodevices relevant to several sectors. The manuscript, presented data, and discussion do not meet the level of Nature Communication at the present stage. Nevertheless, with further work, the manuscript can be reconsidered for publication.

1. The authors claim to have no control over the oscillation dynamics (frequency). This is a rather disappointing statement since a potential reader would expect to see the control of system dynamics. Laser power is a straightforward parameter one can think of, but the author made an explicit statement that indicated precisely the opposite. A qualitative inspection of Figure 4 e and f suggests that increasing the power density from 0.5 to 3.8 W/cm² changes the oscillation frequency of scattering intensity accordingly. I suggest performing experiments for power densities in a forward and backward scan, maintaining the longer measurement time at each power density. One should expect a reversible change in the frequency of scattering intensity oscillations.

2. A fundamental requisite for self-oscillations is a temporal delay between positive and negative feedback mechanisms to avoid a steady state trap. For the oscillation to be feasible in the present system, there should be a delay between reversible plasmon band shift and solvent sorption/desorption in the MOF upper layer. Additional experiments dealing with kinetic analysis of sorption/desorption under temperature change and dynamics of plasmon band shift upon refractive index change should be provided.

3. Phase plot. Authors could estimate the temperature near the plasmonic heat source. That is, it seems that upon changing the amount of sorbed solvent in MOF, the temperature near the plasmonic heater changes because of a scattering band shift. If such a statement is valid, then additional data analysis is needed. For example, one could expect a graphical representation of the relationship between the change of scattering and local temperature.

4. Collective heating. The whole discussion is built on the statement that collective heating of the plasmonic metasurface is negligible. Considering previous works by Baffou (ACSNano 2023 or textbook by the same author), one can argue that collective heating can dominate the temperature that rules solvent dynamics in the MOF layer, especially when the metasurface is subjected to uniform illumination.

5. Figure 2b: what is the difference between the blue and red lines? Labels are needed.

6. Figure 3 d and e: These two subplots show the same data. One of them could be moved to SI.

7. English needs further editing.

Version 1:

Reviewer comments:

Reviewer #1

(Remarks to the Author)

I am happy with the revision. Recommend its publication.

Reviewer #2

(Remarks to the Author)

The manuscript has been appropriately revised and addresses the points raised by the reviewers. The novelty of the manuscript, the self-regulating adsorption and desorption of isopropanol and unique oscillation phenomena by combining adsorption/desorption characteristics in porous MOF thin films with local heating by plasmonic antennas, is high, and it is supported by sufficient theoretical background. I recommend that this manuscript is accepted for publication in Nature Communications in the present form.

Reviewer #3

(Remarks to the Author)

I want to thank the authors I appreciate their efforts to improve the manuscript. The modeling of the oscillations provides

valuable insight into the system's properties and suggests potential future directions for enhancing the oscillator's performance. In principle, I recommend accepting the paper for publication. Before finalizing, I would like to mention four minor points for the Authors to consider:

1. To model the self-oscillatory system, the authors used a simple hysteresis-based model developed from experimental data on the kinetic parameters and hysteresis behaviour of the MOF layer. It is not immediately clear where in the manuscript the hysteresis data for the MOF are located. Presumably, it is Figure 2b. If so, I suggest the authors enhance this figure by adding an inset that clearly shows the hysteresis, helping readers understand the hysteretic behaviour of MOF dynamics from the start.
2. In Figure 6d, the Authors should indicate which lines represent cooling and heating, either by labelling or using arrows on the blue and red lines to show the direction of the hysteresis.
3. In Figures S22 and S23, the arrows on the heating and cooling directions should be indicated.
4. The sections "Discussion" and "Conclusions" should be merged since the text in both sections is somewhat repetitive.

We would like to thank the Reviewers for the time spent to evaluate our work and for their feedback and constructive comments. We did our best to provide additional data and to revise the manuscript according to their comments. Please find hereafter our point-by-point response; in addition the modifications in the manuscript are highlighted in yellow.

Reviewer #1 (Remarks to the Author):

In this manuscript, the authors present an interesting work that integrates Metal-Organic Frameworks (MOFs) with plasmonic metasurfaces to achieve autonomous optical self-regulation. The observed self-oscillatory behavior at the single-antenna level is particularly noteworthy.

RESPONSE: we thank the referee for the positive feedback. Below we address the remaining questions.

However, several issues need to be addressed before the manuscript can be considered for publication. 1. The introduction lacks sufficient discussion of prior research in related fields, particularly in self-adaptation and self-regulation systems. Key information, such as evaluation metrics for self-regulation systems and existing challenges in the research field, is absent. A more thorough comparison with relevant literature would highlight the paper's novelty and provide clearer insights into the current state of research.

RESPONSE: We appreciate the reviewer's insightful comment regarding the comparison with other self-regulating systems. While several such systems have been reported, direct comparisons are not straightforward due to the diverse nature of their self-regulation mechanisms. For example, it is inherently difficult to directly compare a chemo-mechanical negative feedback system with one based on thermo-optical feedback, as the underlying physical principles differ significantly. That said, we have identified and analysed some of the most relevant examples of thermo-optical self-regulating systems, particularly those in which a temperature change induces an optical response that counteracts the initial perturbation.

ACTION TAKEN to address this point: we have added a comparative table in SI (Figure S24) summarizing key thermo-optical self-regulating systems. The table includes information on the materials used, feedback mechanisms, incident light wavelength (when applicable), operational temperature range, corresponding optical response, and thermo-optical sensitivity.

study	materials	mechanism	wavelength of incident light	T ranges	optical change when heating	thermo-optical sensitivity
this work	plasmonic antenna / MOFs	light→heat→ plasmonic shift	852 nm	20 - 26.8°C	0.6 decrease in absorptivity	0,088
Ref 6	Liquid crystal elastomers	light→heat→ actuation	488 nm	22 - 49°C	0.6 decrease in transmittance	0,022
Ref 7	1D photonic crystal / RuO ₂	light→heat→ photonic shift	532 nm	0 - 30°C	0.6 decrease in transmittance	0,020
Ref 8	VO ₂ /SiO ₂ /Au	heat→ emission	-	50 - 82°C	0.24 increase in emissivity	0,008
Ref 9	patterned VO ₂ /SiO ₂ /Al	heat→ emission	-	35 - 80°C	0.48 increase in emissivity	0,011
Ref 10	VO ₂ antennas composite	heat→ emission	-	50 - 65°C	0.56 increase in emissivity	0,037

Furthermore, we have expanded the discussion section to include this discussion, highlighting how our approach fits within and advances the current state of the art:

“The self-regulating performance of our metasurface can be compared to other self-regulating systems reported in the literature. Focusing specifically on devices that are based on thermo-optical negative feedback, such as a homeostatic photonic device,⁴³ liquid crystal elastomers,¹⁹ or materials based on phase transitions⁴⁴ like VO₂,⁴⁵ all these systems are designed to respond to temperature increases with an opposite optical response as summarized in Figure S24. The mechanisms of feedback; the functioning conditions and the type of optical response (absorptivity, emissivity, transmittance) differ among these approaches and each approach comes with its own

advantages and limitations. However, one useful comparison metric is the thermo-optical sensitivity, defined as the ratio between the relative change in the optical property and the corresponding temperature range over which the change occurs. In general one of the main challenge is have this value the highest possible in the temperaure range of interest. In this context, even though a direct comparison is not straightforward, our MOF plasmonic metasurface exhibits a significantly higher sensitivity to temperature changes in the range close to ambient temperature (60% decrease in absorptivity heating from 20 to 26.8°C).

2. Like the negative feedback mechanisms of living systems, the dynamic range in the proposed system have not been explored. For instance, what is the minimum temperature change (sensitivity) that the system can respond to? Additionally, if the temperature exceeds a certain threshold (e.g. maximum laser power), does the system continue to function effectively?

RESPONSE: Thank you for your valuable comment. The reviewer is absolutely right that the self-regulating response of the system deserves deeper discussion. In our case, the thermo-optical sensitivity is not constant across the full temperature range. We performed an additional analysis to evaluate this parameter. This sensitivity can be quantified by calculating the derivative of the scattering evolution at 850 nm with respect to temperature (see Fig S9. a). This is shown in Fig. S9b and c, where two main sensitivity peaks can be identified.

The maximum sensitivity under our experimental conditions occurs between 22 °C and 27 °C, which corresponds to the transition between the Full (F) and MOF-filled (MF) states. In this case the maximum sensitivity is $0.5 \text{ } ^\circ\text{C}^{-1}$ an extremely high value as already discussed in the previous question. A second, less pronounced peak is observed between 80 °C and 100 °C, corresponding to the MF–Empty (E) transition.

These results indicate that the self-regulating effect is optimized to operate most effectively in the 23–27 °C range. However, we emphasize that this response and the range depends on the adsorption/desorption conditions within the porous layer. The system can be readily adapted to extend or shift the sensitivity range by modifying the porosity characteristics, the nature of the vapor, or its concentration (P/P_0) as we will discuss in the next response.

New Figure S9 (a) Evolution of the scattering intensity at 850 nm as a function of the temperature in the presence of 0.7 P/P_0 of IPA for the desorption isobar; (b) and (c) evolution of the sensitivity as function of the temperature for two temperature ranges.

ACTION TAKEN : the new Figure S9 is added in SI. In addition the following paragraphs are added in the manuscript. « *The self-regulating response of the system is not uniform across the entire temperature range, but instead exhibits distinct regions of enhanced sensitivity. This behavior can be analyzed by calculating the derivative of the scattering signal at 850 nm with respect to temperature. As shown in Figure S9b, the highest sensitivity occurs between 22 °C and 27 °C, corresponding to the transition from the Full (F) to the MOF-filled (MF) state. A second, less intense sensitivity peak is observed between 80 °C and 100 °C, associated with the transition from the MF to the Empty (E) state. These findings suggest that the self-regulating effect is most efficient within the 23–27 °C window. Importantly, the sensitivity profile is strongly influenced by the adsorption/desorption dynamics within the porous layer. As a result, the system’s operating range can be readily tuned by altering the porosity characteristics, the chemical nature of the vapor, or the vapor concentration (P/P_0), offering flexibility for application-specific optimization.* »

3. The self-oscillatory behavior and optical self-regulation described in the paper appear to rely on a stable external energy input. However, the long-term stability and sustainability of the system under continuous operation are not discussed. Furthermore, the potential impact of environmental factors, such as temperature fluctuations and humidity changes, on system performance remains unexplored.

RESPONSE: We thank the reviewer for the insightful comment. As correctly noted, the self-oscillatory behavior and optical self-regulation demonstrated in our system currently depend on stable environmental conditions, particularly with respect to temperature and vapor pressure.

The effect of temperature on the refractive index evolution and thermo-optical sensitivity of the MOF layer has been discussed in our previous response. Here, we would like to expand on the influence of vapor pressure, which plays a central role in governing the sorption-desorption equilibrium and, consequently, the dynamic behavior of the system.

To address this point, we conducted additional in situ ellipsometry experiments using an environmental chamber equipped with a heating stage and vapor pressure control. These experiments were designed to measure the temperature-dependent evolution of the refractive index under three different vapor pressure conditions: 0.5, 0.7, and 0.9 P/P₀.

New Figure S10 Evolution of the refractive index of the ZIF-8 layer as function on the temperature for three vapor pressures.

The results clearly show that vapor pressure fluctuations shift the desorption curve: at higher vapor pressure, desorption occurs at higher temperatures, extending the device's sensitivity range to higher temperatures; at lower vapor pressure, the opposite occurs: desorption shifts to lower temperatures shifting the sensitivity range downward. Importantly, this finding not only highlights the need for vapor pressure stability to ensure consistent operation, but also **reveals a valuable opportunity**: by intentionally **tuning the vapor pressure**, it becomes possible to **dynamically adjust the thermal sensitivity range of the device**. This simple principle adds a layer of versatility to the porous system that can't be achieved by other thermo-optical materials (phase changing materials such as VO₂).

ACTION TAKEN : the new Figur S10 is added in SI. The following discussion is added in the main text : « *We explored the possible effect of the vapor pressure. As demonstrated by in situ ellipsometry experiments conducted at different vapor pressures (0.5, 0.7, and 0.9), vapor pressure systematically shift the sorption equilibrium (Figures S10. Specifically, higher vapor pressures delay desorption, thereby shifting the thermo-optical response of the MOF layer toward higher temperatures, while lower vapor pressures produce the opposite effect. This behavior indicates that maintaining a controlled vapor pressure is essential for reliable device operation. At the same time, it presents a significant opportunity: by adjusting the vapor pressure, one can dynamically tune the temperature range over which the device exhibits its large self-regulating response.*»

4. As discussed in Figure 5, “different antennas exhibit a similar frequency but are not at all in phase”, which is somewhat unclear, given that the external stimulus (laser with the same intensity) should be the uniform across all units. Could this discrepancy be attributed to fabrication errors, such as the random distribution of the ZIF-8 colloidal film? A detailed discussion of the phenomenon is necessary, which is crucial for achieving better control over the self-oscillatory fluctuations.

RESPONSE : Thank you for your insightful comment regarding the phase mismatch among different antennas shown in Figure 5. You rightly point out that, under uniform external input, one would expect a more homogeneous response. Following your suggestion, we conducted a more in-depth analysis to evaluate whether local inhomogeneities (fabrication errors ?) and variations in optical conditions could explain the observed variability in oscillatory response. Specifically, we adapted the experiment based on in situ hyperspectral microscopy to perform spectroscopy at the single antenna level of the metasurface covered with ZIF-8 and in the presence of IPA vapor.

This allows mapping not only the scattering response at a single wavelength (as in the first version of the manuscript) but also the « optical homogeneity » between antennas.

Our findings, now reported in the revised manuscript, indicate that while bare antennas exhibit homogeneous optical behavior (Figure 2), the presence of ZIF-8 and vapor induces significant variation in the scattering response of neighbor antennas. This is now shown in this new Figure S16 as follows.

New Figure S16 (a) Scheme of the array, (b) scattering intensity curve of a row of antenna/ZIF-8 in presence of vapor. (c) selected scattering spectra of antenna/ZIF-8s. (d) Evolution of the scattering intensity at 850 nm for a row of antenna/ZIF-8s.

This new figure reports the optical response of individual antennas on a single line (Figure S16a) in presence of ZIF-8 colloidal film and vapor. The corresponding scattering spectra from multiple antennas along the line, shown in Figure 5c, reveal that the scattering response is not homogeneous. A selection of different antennas is presented in Figure S16c. The red vertical line in panels (c) and (d) indicates the 850 nm excitation wavelength, highlighting how variations in scattering near this wavelength can influence local absorption and heating. Figure S16(d) illustrates the spatial distribution of scattering intensity at 850 nm, measured from several antennas across the line. The variation in intensity values suggests non-uniform optical behavior among the antennas.

This supports the Reviewer's hypothesis that the random distribution of ZIF-8 in the composite material is responsible for the observed variability in dynamic response. Indeed, the ZIF-8 film consists of randomly stacked colloidal particles, leading to non-uniform contact with each antenna. During capillary condensation, the local distribution of ZIF-8 particles and IPA liquid becomes further inhomogeneous across the sample. As shown in Figure 3, the antennas are particularly sensitive to refractive index changes at their extremities. Therefore, the non-uniform distribution of ZIF-8 and condensed vapor likely leads to antenna-specific variations in absorption at 850 nm (the laser wavelength), which in turn results in different local heating and oscillatory behavior.

This additional analysis allows us to better explain why the antenna are not oscillating in phase. Additionally, **this unintended variability is actually an advantage** to support our approach, as (i) it provides an additional evidence that the oscillations are not due to artifacts and (ii) it suggests that, by tuning the sorption and optical properties, different oscillatory behaviors can be obtained.

A more detailed dynamic analysis, supported by numerical simulations, is provided in our response to Reviewer 3.

ACTION TAKEN: The analysis and the new Figure S16 is added in SI, the discussion is added in the main manuscript " To further investigated the possible origin of these different oscillatory behavior, we performed a spectroscopic mapping at the single antenna level of the metasurface covered with ZIF-8 and in the presence of IPA vapor. While bare antennas exhibit homogeneous optical behavior (Figure 2), the presence of ZIF-8 and vapor induces a certain variation in the scattering response of neighboring antennas (Figure S16). The ZIF-8 film consists of randomly stacked colloidal particles. Given that the antennas are particularly sensitive to refractive index changes at their extremities (as shown in Figure 3), the non-uniform distribution of ZIF-8 and condensed vapor results in variations in absorption at 850 nm and different local heating that is responsible for the oscillatory process. This unintended variability can be seen as an advantage to support our approach as it provides an additional evidence that the oscillations are not due to artifacts. For instance, possible fluctuations in vapor pressure or in incident light

intensity would result in a simultaneous variation of the scattering intensity for all the antennas. Instead, these results indicate that each individual ZIF-8/antenna composite behaves as an individual oscillator, as illustrated in Figure 5(d). These observations also indicate that by controlling the optical properties, different oscillatory behaviors can be obtained. »

5. More specific applications can be discussed, for instance, optical manipulation (Appl. Phys. Rev. 9, 031303 (2022)), information processing (Nature Communications 15 (1), 9658, (2024)), and others.

RESPONSE : we thank the reviewer for the suggestion, the references have been added.

6. It's recommended that the authors pay more attention to enhancing the quality of the figures. The text is currently too small and difficult to read, and the overall aesthetics of the figures could be improved. These issues should be addressed to enhance the readability and clarity of the graphical content.

RESPONSE : following the Reviewer's suggestion we have modified some Figures (Figure 3, 4, 5 and 6) to improve their readability and clarity.

Reviewer #2 (Remarks to the Author):

In this manuscript, the authors report pioneering research on self-regulating adsorption and desorption of isopropanol and unique oscillation phenomena by combining adsorption/desorption characteristics in porous MOF thin films with local heating by plasmonic antennas. The oscillation phenomenon that occurs due to the coupling of changes in the plasmon resonance frequency caused by changes in the dielectric constant accompanying the adsorption and desorption of molecules (isopropanol) exhibit high novelty and is expected to be developed for various applications in the future. Since an important and highly novel discovery has been reported, it is appropriate to accept the paper after addressing the comments below.

RESPONSE : we thank the reviewer for recognizing the novelty of our work.

The introduction contains almost no background information on the MOF-Plasmonic array system used in this study. It is requested to add that the information on previous research using the MOF (porous materials)-plasmonic antenna system and the circumstances that led to the concept of this study.

RESPONSE : We thank the reviewer for the recommendation. Indeed there is a relevant state of the art on so called « MOF hybrid plasmonic materials » that needs to be discussed.

ACTION TAKEN :

- 1) To clearly position this work in the state of the art, we introduce the term « hybrid » when referring to our materials in the title, and in the text.
- 2) The following sentence is added in the text « *MOF hybrid plasmonic materials have been previously used for sensing, nanomedicine or catalytic applications as summarized in Table 1 in SI. In this study, we introduce a previously unexplored and unique functionality of MOF-hybrid plasmonic materials...* »
- 3) We provide in SI a table describing the state of the art

State of the art on MOF hybrid plasmonic materials

In recent years, MOF hybrid plasmonic materials have emerged as multifunctional platforms combining the sorption properties of MOFs with the field-enhancing capabilities of metallic nanostructures. These hybrid architectures have shown promise in various domains, including photocatalysis, drug delivery, and ultra-sensitive molecular sensing (e.g., SERS). Their performance depends critically on the nanostructural design, including core-shell geometries, interface engineering, and the type of MOF and plasmonic metal used.

The table below illustrates several key examples from the literature that highlight the diversity of MOF hybrid plasmonic nanostructures and their applications in sensing, catalysis, and therapy:

Publication	Type of MOF	Plasmonic Nanoparticles	Structure	Applications
Dhakshinamoorthy et al. (2020), ChemMedChem	-	Au NPs	Au NPs decorated MOF	Photodynamic therapy, drug

				delivery, anticancer
Muhamed et al. (2023), Inorg. Chem.	NU-1000 (Zr-based)	Au NPs	Au NPs grown on NU-1000	Hydrogen evolution, photochemical and electrochemical catalysis
Huang et al. (2020), Chem. Phys. Lett.	ZIF-8	Ag nanowires	Core-shell Ag@ZIF-8 nanowires	Pb ²⁺ sensing
Liu et al. (2015), Adv. Mater.	ZIF-8	Ag nanowires	Core-shell Ag@ZIF-8 nanowires	Solar-driven butanol separation
Chen et al. (2025), Small Methods	MOF-801 (Zr-based)	Plasmonic nanocrystals	MOF on plasmonic crystal composite	CO ₂ photoreduction
Sun et al. (2025), Sens. Act. B	MIL-101	Ag NPs	MIL-101@Ag with PDA layer	SERS enhancement
Li et al. (2018), Nano Res.	ZIF-8	Au nanorods	Au NRs encapsulated in ZIF-8	pH-triggered drug release and photothermal therapy
Li et al. (2019), J. Mater. Chem. A	HKUST-1 (Cu)	Ag NPs	Core-shell HKUST-1@Ag on SPCE	SERS detection of PAHs and 4-ATP

- It is reported in the manuscript that the temperature change due to plasmonic heating is 6.3 degree C. The temperature change seems to be very small. It is necessary to show that the temperature dependence of the transition between the F-MF-E states of MOF (not just the relative temperature change) occurs under the temperature change estimated for the plasmonic heating in the experimental environment.

RESPONSE : Thank you for your valuable comment. As correctly pointed out, the temperature increase due to plasmonic heating is estimated to be 6.8 °C. The evolution of the optical properties of the MOF as a function of temperature is shown in Figure 2b, where it is evident that this temperature range (from approximately 20 °C to 27 °C) corresponds to the transition from the fully loaded (F) state to the medium-filled (MF) state.

We agree that achieving the fully empty (E) state would require a significantly higher temperature, up to around 110 °C. However, under our experimental conditions, the system does not reach such high temperatures for two main reasons: (1) the incident laser power is limited, and (2) most importantly, this reflects the core of our approach, the device is self-regulating, meaning that as the laser power increases, the material absorbs less light due to its intrinsic optical response, thus preventing further heating.

ACTION TAKEN : the following text is added : « *Under our irradiation conditions (up to a maximum 3.9 W cm⁻²), the system transitions from the (F) state to the (MF) state. Higher temperatures required to reach the fully empty (E) state (up to ~110 °C) are not achieved under our experimental conditions, due to both the limited laser power and the self-regulating nature of the device, which reduces light absorption at higher intensities* »

- A quantitative description of the oscillation frequency of plasmon scattering is required (around 50 s?), and additional information is needed on the factors that determine the oscillation frequency and the possibility of modulation.

RESPONSE : Thank you for your insightful comment. The oscillation frequency is determined by several factors, the most important being the characteristic time of vapor uptake, which is on the order of tens of seconds and represents the « slowest » process. This aspect is discussed in the response of a comment/suggestion of Reviewer 3 concerning the temporal delay. Additionally, the presence and nature of the hysteresis in the adsorption/desorption process also play a crucial role in defining the frequency and its modulation.

ACTION TAKEN : This aspect is now thoroughly discussed in Section 2.4 where we include kinetic experiments, simulations that provide a quantitative description of the oscillatory behavior, as illustrated in the new Figure 6.

Reviewer #3 (Remarks to the Author):

The paper by Amyar and coworkers deals with developing a self-oscillatory system based on thermoplasmonics

and MOF components. The authors have proposed a conceptually fresh framework with the potential for fundamental studies of non-linear nanosystems and for developing nanodevices relevant to several sectors. The manuscript, presented data, and discussion do not meet the level of Nature Communication at the present stage. Nevertheless, with further work, the manuscript can be reconsidered for publication.

We sincerely thank the reviewer for their thoughtful feedback and constructive suggestions.

We acknowledge that further clarification and quantitative support were needed, particularly regarding the oscillation dynamics. Since comments 1 and 2 are closely related, we have provided a combined and detailed response addressing both aspects adding new kinetic analysis and simulations to strengthen the scientific discussion.

1. The authors claim to have no control over the oscillation dynamics (frequency). This is a rather disappointing statement since a potential reader would expect to see the control of system dynamics. Laser power is a straightforward parameter one can think of, but the author made an explicit statement that indicated precisely the opposite. A qualitative inspection of Figure 4 e and f suggests that increasing the power density from 0.5 to 3.8 W/cm² changes the oscillation frequency of scattering intensity accordingly. I suggest performing experiments for power densities in a forward and backward scan, maintaining the longer measurement time at each power density. One should expect a reversible change in the frequency of scattering intensity oscillations.

2. A fundamental requisite for self-oscillations is a temporal delay between positive and negative feedback mechanisms to avoid a steady state trap. For the oscillation to be feasible in the present system, there should be a delay between reversible plasmon band shift and solvent sorption/desorption in the MOF upper layer. Additional experiments dealing with kinetic analysis of sorption/desorption under temperature change and dynamics of plasmon band shift upon refractive index change should be provided.

RESPONSE : We thank the reviewer for these insightful and important comments.

Indeed our statement about 'no control over the oscillation dynamics' was not accurate. We carried out a further analysis of our experimental data and confirm that, as indicated by the reviewer and shown in (the previous) Figure 4 (e), an increase in oscillation frequency can be observed with increasing laser intensity for several antennas. As indicated in the response of Reviewer 1, from the new experiments, we realized also that the variability in oscillatory response can be explained by the local heterogeneities in ZIF-8 packing.

To achieve control, understanding the origin of the oscillation dynamics is essential. We also acknowledge the importance of identifying the temporal delay and kinetic conditions that give rise to self-oscillations to ultimately control them.

ACTION TAKEN

The following experiments and analysis have been made :

1) **Kinetic analysis**

We have performed new experiments of both plasmonic heating rate and adsorption/desorption rate in the MOF layer, which allowed us to estimate the characteristic timescales of the involved processes. These measurements confirm the presence of a significant **temporal delay** between the fast plasmonic heating and the slower adsorption/desorption response, which is further influenced by the intrinsic **hysteresis** of the MOF.

In the first experiment, we focused on the plasmonic heating rate. We simulated the transient thermal behavior of plasmonic gold nanoantennas subjected to localized photothermal excitation. The heat dissipation dynamics were modeled using a lumped capacitance approach, assuming that the temperature within the gold volume remains spatially uniform during heating due to its high thermal conductivity.

The thermal time constant τ of the gold structure was calculated as:

$$\tau = \frac{\rho C_p V}{h A}$$

where ρ is the mass density of gold, C_p its specific heat capacity, V its volume, A the effective heat exchange surface area, and h the convective heat transfer coefficient. The absorbed power was inferred from the maximum temperature rise experimentally observed, and the temperature evolution over time was fitted using an exponential function of the form:

$$T(t) = T_{max}(1 - e^{-t/\tau})$$

Where T_{max} is the experimentally measured maximum temperature increase. In this case, the plasmonic heating is very fast with a **time constant of 63 μ s**.

New Figure S19 Evolution of the temperature of the plasmonic antenna upon light irradiation

The second experiment aimed at evaluating the characteristic sorption time constant of the MOF. For this purpose, we fabricated a simplified sample consisting of a glass substrate coated with spherical gold nanoparticles on 2 cm^2 , chosen for their scalability and ease of use (as shown in *Nat Commun* **15**, 1156 (2024) and *Adv. Optical Mater.* **15** (2025), 2500079). In this context, we only required a comparable heat source, and spherical nanoparticles provided a practical alternative, since the fabrication of plasmonic nanoantennas is less easily transferable and would require advanced techniques such as electron beam lithography.

The plasmonic nanoparticles were coated with the MOF layer, and we monitored the dynamic evolution of the MOF's refractive index over time upon laser excitation (532 nm). As the laser is applied, the refractive index of the MOF decreases, indicating that desorption occurs due to local heating. For clarity, we present the absolute value of the change in refractive index versus time.

From this response, we were able to extract a characteristic time constant of **approximately 20 seconds**, which give an indication the sorption kinetics under plasmon-induced thermal stimulus.

New Figure S20 Sorption kinetics : (absolute value) variation of the refractive index of the MOF layer upon light irradiation measured by ellipsometry

These findings indicate that the dominant time-limiting factor responsible for the self-oscillation delay is the sorption time constant of the vapor in/from the MOF layer.

2) Dynamic simulation

To study the oscillatory dynamics the reviewer suggested performing longer experiments at variable light intensity. Despite our efforts, carrying out longer experiments was experimentally challenging at such a high magnification, as it requires maintaining focus and avoiding drift or vapor pressure drops. In addition, the antenna samples were partially damaged after multiple runs and cannot be refabricated exactly in the same way. Producing an identical system is challenging, time-consuming, and costly, since fabrication requires state-of-the-art electron beam lithography facilities.

Nevertheless, to answer to the reviewer's questions, we carried out **dynamic simulations** to describe the oscillatory dynamics, that provide an even more complete understanding on the system. Based on experimental kinetic parameters and hysteresis behavior of the MOF layer, we developed a simple **hysteresis-based model** that reproduces the **self-oscillatory behavior**. This model allows us to systematically explore the influence of various parameters, in particular laser intensity and hysteresis shape, on the oscillation dynamics.

We considered our system as a **material whose optical absorptivity depends on temperature, with a hysteretic response and a finite time constant (sorption time constant) governing its evolution**. In particular, the absorptivity exhibits different transition thresholds during heating and cooling due to the hysteresis of the F to MF states reported in Figures 2.

Physical Model : The system under consideration consists of a material with an initial absorptivity, which absorbs optical power from a constant incident optical flux. The temperature of the material changes as a result of both the absorption of incident optical power and thermal dissipation to the environment. The material is assumed to be in thermal equilibrium with the surroundings, and the thermal behavior of the system is modeled using differential equations derived from energy conservation principles.

Governing Equations : The thermal behavior of the system is modeled by two coupled differential equations: one for the temperature, $T(t)$ and one for the absorptivity, $\alpha(t)$. These equations describe the evolution of temperature and absorptivity over time based on the incident power, the thermal dissipation, and the feedback between temperature and absorptivity.

The rate of change of temperature, dT/dt is given by the following equation derived from energy conservation:

$$\frac{dT}{dt} = \frac{\alpha(t)P_{in} - k(T - T_{env})}{C}$$

where:

- P_{in} is the incident optical power,
- $\alpha(t)$ is the time-dependent absorptivity of the material.
- k is the thermal dissipation coefficient,
- T_{env} is the ambient temperature
- C is the heat capacity of the material.

The absorptivity, $\alpha(t)$, evolves over time as a function of temperature. A sigmoid function is used to model the temperature-dependent change in absorptivity with different transition thresholds and steepness for heating and cooling phases. The absorptivity evolves with a delayed relaxation toward this equilibrium value with a characteristic time constant τ (20s as determined experimentally).

$$\frac{d\alpha}{dt} = \frac{\alpha_{eq}(T, \text{mode}) - \alpha}{\tau_\alpha}$$

where mode switches between heating and cooling depending on the sign of the temperature derivative, enforcing the hysteresis. The absorptivity follows different paths during heating and cooling phases, with transitions occurring at specific threshold temperatures.

- Heating Phase: When the system is in the heating mode, the absorptivity increases as the temperature rises.
- Cooling Phase: When the system switches to cooling mode (due to a decrease in temperature), the absorptivity decreases as the temperature drops.

The temperature dependence of absorptivity was modeled using sigmoidal functions to capture the smooth yet sharp transition between low and high absorptivity states near threshold temperatures. These functions provide a convenient phenomenological description of hysteresis by defining separate transition branches for heating and cooling.

For each mode (heating or cooling), the equilibrium absorptivity is given by:

$$\alpha_{eq}(T) = \alpha_{min} + \frac{\alpha_{max} - \alpha_{min}}{1 + \exp[\beta(T - T_{th})]}$$

where:

- α min and max are the lower and upper bounds of absorptivity.
- T_{th} is the threshold temperature at which the transition occurs.
- β is a steepness parameter controlling how sharply the transition happens.

New Figure S21 Absorptivity vs temperature hysteresis curve modeled using sigmoidal functions

As shown in the Figure above, for the simulation we used values transition thresholds, β , α min and max and time constant determined experimentally. For the other parameter values (heat capacity, dissipation coefficient, incident power) we imposed physically reasonable values to explain the observed experimental behavior. The ultimate goal is to propose a mechanism and gain indications about the influence of various parameters.

The results are shown in the new Figure 6 (d-g) and in SI.

New Figure 6

As shown in Figure 6(e), the analysis displays an oscillatory behavior in absorptivity with a period of around 45 seconds, which is comparable to our experimental observations. The system demonstrates absorptivity oscillations that results from an elliptical loop in the hysteresis curve, capturing the delayed transitions between heating and cooling states that drive the self-oscillatory behavior. The shape and path of this hysteretic loop has an thus impact the frequency of the oscillation. To further explore that, we performed the same analysis using hysteresis curves with identical threshold temperatures but varying steepness parameters (β). This variation can be describe porous networks with different pore size distribution: monodisperse pores lead to sharper transitions, while a polydisperse distribution results in broader, less abrupt changes. (Ceratti et al Nanoscale 2015). As shown in Figure 6(g), the frequency of the oscillation increasing for shaper hysteresis. Similarly, as shown in Figure S22, broader hysteresis results in a larger loop and a decrease of the frequency . No oscillations are observed for no hysteresis or very narrow hysteresis, suggesting the importance of the presence of the hysteresis.

New Figure S22 Oscillatory response for hysteresis with different temperature threshold. Broader hysteresis results in a larger loop and a decrease of the frequency (no oscillation are observed for no hysteresis or very narrow hysteresis).

This simple analysis suggests that by varying the sorption properties and optical response of the porous media, different oscillatory behaviors can be achieved. This may further explain why heterogeneous packing of ZIF-8 on the antenna leads to variations in oscillatory behavior as discussed in Figure 5 and the response to Reviewer 1. Finally, the effect of light intensity was examined. As pointed out by the reviewer, observations on selected antennas (Figure 6(b), indicates that increasing the light intensity results in a higher oscillation frequency. We performed simulations with increasing light intensity, as shown in new Figure S23. Interestingly, we confirmed that the oscillation period decreases with light intensity until the absorptivity reaches a low value, where the steepness of the hysteresis decreases, causing the oscillations to slow down and then stop.

New Figure S23 Effect of light intensity. Simulations with increasing light intensity (0.02 increment every 400s). The oscillation period decreases with light intensity until the absorptivity reaches a low value, where the steepness of the hysteresis decreases, causing the oscillations to slow down and then stop.

MODIFICATIONS ON THE MANUSCRIPT: all the data and the technical discussion provided above are now included in the manuscript (new Figure 6) and in SI. We also provided a new section in the manuscript “5. Discussion” to discuss the oscillatory behaviour (based on Figure 6) by providing insights and perspectives on how to control the oscillatory behaviour.

3. Phase plot. Authors could estimate the temperature near the plasmonic heat source. That is, it seems that upon changing the amount of sorbed solvent in MOF, the temperature near the plasmonic heater changes because of a scattering band shift. If such a statement is valid, then additional data analysis is needed. For example, one could expect a graphical representation of the relationship between the change of scattering and local temperature.

RESPONSE : thank you for the suggestion. The evolution of the scattering intensity (at 850 nm) as fonction of the temperature is reported hereafter.

New Figure S9 (a) Evolution of the scattering intensity at 850 nm as a function of the temperature in the presence of 0.7 P/P₀ of IPA for the desorption isobar

ACTION TAKEN : the plot is now added in SI and referred in the main text when discussing the thermal sensitivity of the metasurface.

4. Collective heating. The whole discussion is built on the statement that collective heating of the plasmonic metasurface is negligible. Considering previous works by Baffou (ACSNano 2023 or textbook by the same author), one can argue that collective heating can dominate the temperature that rules solvent dynamics in the MOF layer, especially when the metasurface is subjected to uniform illumination.

RESPONSE : We appreciate the remark that is important. We believe the Reviewer is referring to the seminal work by Baffou et al., ACS Nano 2013, *Photoinduced Heating of Nanoparticle Arrays*, which we cited in the first version of the manuscript as reference 39. Our findings are consistent with Baffou's work. As shown in Figure 2(b) of the ACS Nano 2013 paper, plasmonic particles spaced 500 nm apart behave as individual hot spots, with very little temperature increase between them. In our case, the antennas are spaced 1 μm apart, an even larger distance, which explains the very low collective heating effect that we determined by thermal simulation (Figure 4).

Figure 2 from Baffou et ACS Nano 2013

However, we agree that our initial statement claiming that collective heating is negligible may not be entirely accurate. To better support this discussion, we performed additional thermal simulations by varying the distance between the antennas.

New Figure S18 Heat map and line plots of temperature elevation for two nanoantenna placed at different distances

As shown in the new Figure S18, reducing the antenna spacing to 250nm leads to an increase in global heating (due to the higher density of photoheating particles) and enhanced temperature in the regions between antennas, indicating that collective heating is indeed favored at shorter distances. These results support our hypothesis and suggest that, in perspective, denser arrays of antennas can heat more efficiently, exhibit collective thermal effects, and potentially enable synchronization phenomena.

ACTION TAKEN : the plot is now added in SI and discussed in the main text. We toned down the « no thermal coupling » statement by replacing with '*the temperature rise is located in the close vicinity of the antenna with a very low heating between the antenna*'. In addition we discussed the thermal coupling in the discussion section to open perspective for achieving a collective oscillatory response.

5. Figure 2b: what is the difference between the blue and red lines? Labels are needed.

RESPONSE : Thank you for spotting this. The curves refer to heating and cooling processes

ACTION TAKEN : the label was added in Figure 2b

6. Figure 3 d and e: These two subplots show the same data. One of them could be moved to SI.

RESPONSE : Thank you, yes indeed this is the same set of data. We remove the previous Figure 3e from the new Figure 3 that was simplified to improve the clarity (as requested by Reviewer 1).

7. English needs further editing.

RESPONSE : the text was doublechecked and revised to improve the English and correct the mistakes.

We would like to thank the Reviewers once again for their comments and suggestions. Please find hereafter our response.

Reviewer #1 (Remarks to the Author):

I am happy with the revision. Recommend its publication.

Response : We would like to thank again the reviewer.

Reviewer #2 (Remarks to the Author):

The manuscript has been appropriately revised and addresses the points raised by the reviewers. The novelty of the manuscript, the self-regulating adsorption and desorption of isopropanol and unique oscillation phenomena by combining adsorption/desorption characteristics in porous MOF thin films with local heating by plasmonic antennas, is high, and it is supported by sufficient theoretical background. I recommend that this manuscript is accepted for publication in Nature Communications in the present form.

Response : We would like to thank again the reviewer.

Reviewer #3 (Remarks to the Author):

I want to thank the authors I appreciate their efforts to improve the manuscript. The modeling of the oscillations provides valuable insight into the system's properties and suggests potential future directions for enhancing the oscillator's performance. In principle, I recommend accepting the paper for publication.

Response : We would like to thank again the reviewer and the for the suggestions.

Before finalizing, I would like to mention four minor points for the Authors to consider:

1. To model the self-oscillatory system, the authors used a simple hysteresis-based model developed from experimental data on the kinetic parameters and hysteresis behaviour of the MOF layer. It is not immediately clear where in the manuscript the hysteresis data for the MOF are located. Presumably, it is Figure 2b. If so, I suggest the authors enhance this figure by adding an inset that clearly shows the hysteresis, helping readers understand the hysteretic behaviour of MOF dynamics from the start.

Response : thank you for the suggestion. The inset have been added in Figure 2(b) and refered in the text from the begining of the discussion.

2. In Figure 6d, the Authors should indicate which lines represent cooling and heating, either by labelling or using arrows on the blue and red lines to show the direction of the hysteresis.

Response : the labels have been added

3. In Figures S22 and S23, the arrows on the heating and cooling directions should be indicated.

Response : the arrows have been added

4. The sections "Discussion" and "Conclusions" should be merged since the text in both sections is somewhat repetitive.

Response : according to the journal's guidelines only the discussion section is included now.